# Host-driven temperature dependence of Deformed wing virus infection in honey bee pupae

Evan C. Palmer-Young [1✉], Eugene V. Ryabov[1,2], Lindsey M. Markowitz[1,3], Dawn L. Boncristiani[1], Kyle Grubbs[1], Asha Pawar[1], Raymond Peterson[1] & Jay D. Evans[1]

The temperature dependence of infection reflects changes in performance of parasites and hosts. High temperatures often mitigate infection by favoring heat-tolerant hosts over heat-sensitive parasites. Honey bees exhibit endothermic thermoregulation—rare among insects—that can favor resistance to parasites. However, viruses are heavily host-dependent, suggesting that viral infection could be supported—not threatened—by optimum host function. To understand how temperature-driven changes in performance of viruses and hosts shape infection, we compared the temperature dependence of isolated viral enzyme activity, three honey bee traits, and infection of honey bee pupae. Viral enzyme activity varied <2-fold over a > 30 °C interval spanning temperatures typical of ectothermic insects and honey bees. In contrast, honey bee performance peaked at high ($\geq$ 35 °C) temperatures and was highly temperature-sensitive. Although these results suggested that increasing temperature would favor hosts over viruses, the temperature dependence of pupal infection matched that of pupal development, falling only near pupae's upper thermal limits. Our results reflect the host-dependent nature of viruses, suggesting that infection is accelerated—not curtailed—by optimum host function, contradicting predictions based on relative performance of parasites and hosts, and suggesting tradeoffs between infection resistance and host survival that limit the viability of bee 'fever'.

[1] USDA-ARS Bee Research Laboratory, Beltsville, MD, USA. [2] Department of Entomology, University of Maryland, College Park, MD, USA. [3] Department of Biology, University of Maryland, College Park, MD, USA. ✉email: ecp52@cornell.edu

Temperature is a fundamental driver of biological rates and a strong predictor of infection outcome in many systems where host temperature is variable[1,2]. Metabolic theory provides a framework for explaining the temperature dependence of physiological and ecological phenomena, including host-parasite interactions[3–5]. This approach uses models derived from enzyme kinetics to describe the temperature dependence of organismal performance (i.e., the 'thermal performance curve')[6,7]. The 'thermal mismatch hypothesis' postulates that the outcome of antagonistic bipartite interactions reflects the relative performance of the interacting parties[8]. Hence, temperature is predicted to impact infection when hosts and parasites respond differently, or in extreme cases, oppositely, over a given temperature range, with parasite growth maximized at temperatures where parasites outperform hosts and minimized where hosts outperform parasites[4,9,10]. In general, hosts are expected to be most resistant to infection at temperatures within the preferred host range, resulting in lower infection of warm-adapted hosts at warm temperatures. In diverse endo- and ectothermic plants and animals, including insects[11–13], infection can be reduced by high body temperatures (i.e., fever[12,14–16]) that compromise the essential functions of parasites and/or potentiate the immune function of hosts[17–19].

The thermal biology of honey bees offers a unique opportunity to study the effects of body temperature on infection. These social bees are endothermic at both the individual and colony level[20–22], expending considerable energy to maintain brood temperatures within a remarkably narrow, 34–36 °C range close to mammalian body temperature[20,21,23]. Honey bee physiology is also highly temperature-sensitive, with peak function achieved within a narrow range of high temperatures. Adults are unable to fly at body temperatures below 30 °C, with muscle force peaking at 38 °C[24], and resting metabolic rate increasing 5-fold between 20 and 35 °C[25]. The immature life stages, which develop entirely within the thermoregulated colony core, are even more temperature-sensitive than are adults. Successful development occurs only between 29 and 37 °C, with mortality and developmental defects occurring outside of the 32–36 °C region and optimal brain development only at 33.5–35 °C[26]. Nevertheless, honey bees can experience temperatures as low as 5 °C in broodless winter colonies[23]—although the 20–37 °C range is more typical—and tolerate temperatures of >40 °C during flight[22].

Honey bee body temperature has the potential to affect infections by a well-documented assortment of parasites and pathogens, which can adversely affect individual and colony health. These include invertebrate animals, eukaryotic fungi and protozoa, prokaryotic bacteria, and a suite of viruses, many of which also infect less endothermic species[27,28]. High temperatures achieved by colonies during the brood-rearing season can decrease infection with *Varroa* mites[29], *Ascosphaera apis*[30], *Nosema apis* and *N. ceranae*[31], and viruses[32–34]. These results are consistent with mismatched responses to temperature between parasites adapted to diverse ectothermic hosts and endothermic honey bees, which appear to gain a relative advantage over these parasites as temperatures increase towards the peak performance range of bees.

One parasite that has emerged as a formidable honey bee antagonist is Deformed wing virus (DWV). This RNA virus is likely the most prevalent virus among managed bees worldwide[35,36]. Although historically DWV has not been a major threat to bee health, its virulence has been augmented by the global spread of the *Varroa destructor* mite, which readily transmits the virus among developing pupae and adult bees, resulting in adults with wing deformities and drastically reduced performance and longevity[35]. Outside of honey bees, the virus has

been found in eight orders of arthropods, including a diversity of wild bees in which its pathogenicity and spillover from honey bees is of potential concern[37,38]. Yet no antiviral treatments currently exist for honey bee colonies, motivating studies of virus traits, bee defenses, and environmental factors—including temperature—that affect infection[39]. Adult bees reared at 37 °C show reduced DWV proliferation but also dramatically higher host mortality relative to lower temperatures[33]. However, effects of temperature on viral infection of susceptible pupae, and the relative contributions of hosts and viruses to such effects, remain unknown. Although more stable than that of honey bee adults, temperature in this life stage can vary by several degrees temporally across the season[20], spatially within the brood cluster[40], and between female (worker) and male (drone) brood, the latter of which are preferred by virus-vectoring mites;[40,41] much greater temperature ranges occur in DWV-associated, non-honey bee host species that do not thermoregulate during development.

The characterization of isolated parasite enzymes offers an opportunity to understand the intrinsic traits of unculturable viruses outside of hosts, while the use of reporter-tagged virus clones allows for controlled inoculations with traceable parasites that can be distinguished from preexisting infections. Both approaches have recently been applied to DWV[42,43]. Initial evaluation of the first cloned virus enzyme—a protease that cleaves the whole-genome viral polyprotein into individual proteins—showed less than 20% variation in rates of activity across a > 20 °C temperature range[43], suggesting that the virus itself is relatively temperature-insensitive. To evaluate the effects of temperature on host-parasite interactions in virus-infected, developing honey bees and the extent to which these dynamics are explained by mismatches between the thermal performance curves of hosts and parasites, we modeled and compared the temperature dependence of performance between a cloned viral protease, three honey bee traits, and pupal infection with a luciferase-tagged clone of DWV.

The preliminary evidence for a contrast between the temperature sensitivity of honey bee physiology—which is optimized at relatively high temperatures—with the relatively constant function of a key viral enzyme across a wide temperature range suggested three alternative hypotheses with divergent predictions for the effects of temperature on infection across the temperature range of the colony: First, if infection is governed primarily by the activity of virus enzymes, then a temperature increase from that typical of an ectothermic insect host to that of an endothermic honey bee colony should have weak effects on infection, reflecting low temperature sensitivity of the virus. Second, if infection is limited by efficient function of the host immune system, then as temperature approaches the high levels over which honey bee metabolism is optimized, infection should rapidly decrease. Third, if rapid proliferation of the virus is instead dependent on strong host performance, then as temperature rises into the range of optimum honey bee function, infection should increase. This differs from the second hypothesis in that the virus is dependent on optimal host function, rather than limited by it, such that infection correlates directly (rather than inversely) with host performance.

## Results

**Temperature dependence of viral protease activity and honey bee traits**. Relative to the bee traits, activity of the viral protease peaked at lower temperatures but was overall less temperature-sensitive. The protease's broad peak of activity between 15 and 30 °C contrasted with the narrow, 35–40 °C peak for each bee life stage (Fig. 1). The protease's estimated temperature of 50% inactivation $T_h$ (estimate 30.2 ± 2.6 °C SE) was at least 7 °C lower than that of any of the bee traits (40.7 ± 1.8 °C for larval

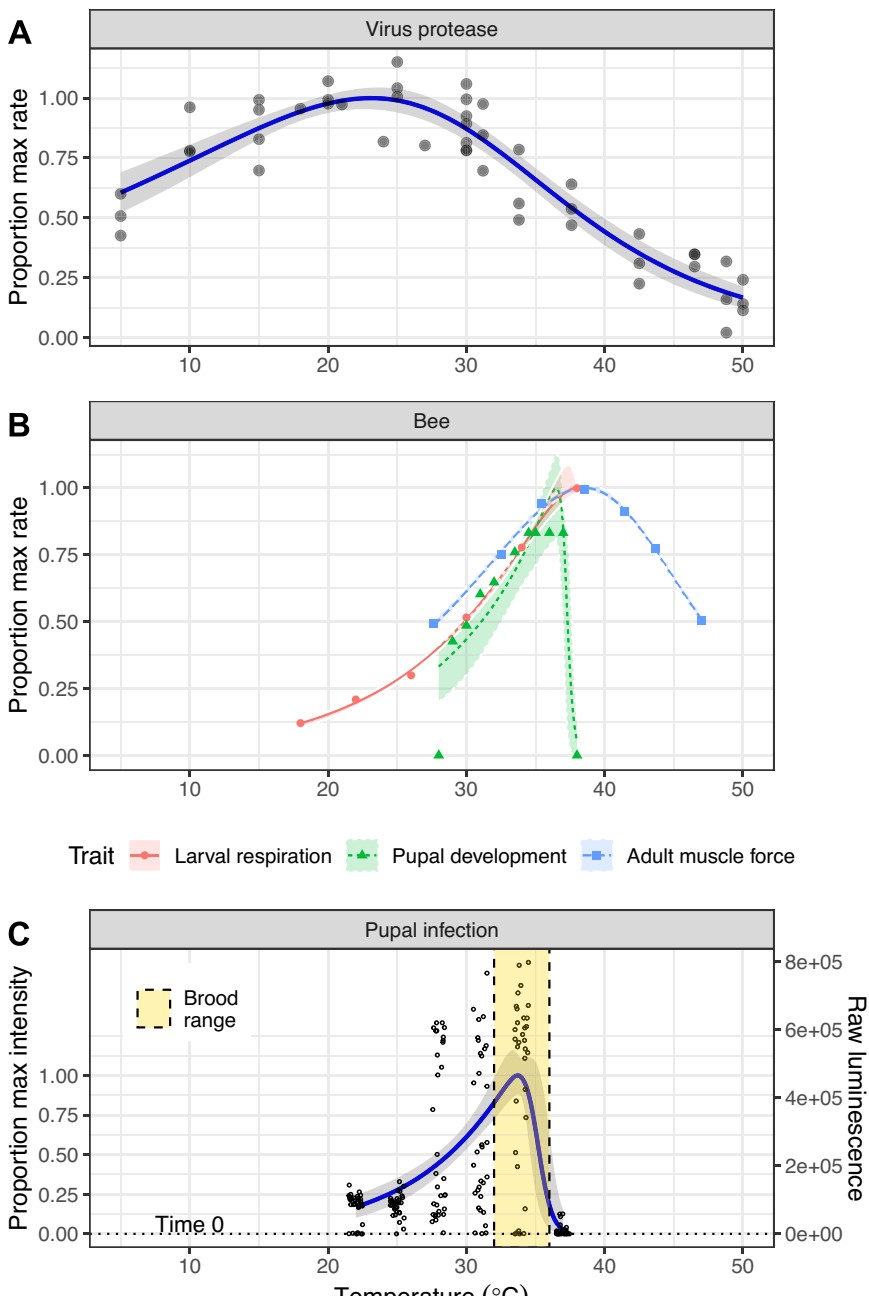

**Fig. 1 Effects of temperature on Deformed wing virus, honey bees, and infection.** Panels show data (points) and model predictions (lines), and 95% bootstrap confidence intervals (shaded bands) for virus protease activity (**A**, upper panel), three bee traits (**B**, middle panel), and pupal infection (**C**, lower panel). Data for larval respiration (red circles and solid line), pupal development (green triangles and dotted line), adult muscle force production (blue squares and dashed line), and brood temperature range (yellow vertical region in infection panel) taken from previous observations[20,24,26,45]. In the lower panel, the secondary y-axis indicates the raw sample luminescence (i.e., before transformation to proportion of maximum intensity) and the horizontal line dotted line represents the mean luminescence of $n = 16$ samples frozen immediately post-injection. To provide a comparable scale for the different measurements, raw values for all traits are shown as proportions of the trait's peak model-predicted value.

respiration, $37.2 \pm 0.58\,°C$ for pupal development, and $40.8 \pm 0.62\,°C$ for adult muscle force production (Fig. 2)). The protease's activation energy $e$ ($0.30 \pm 0.074\,eV$), a measure of temperature sensitivity, was also more than two-thirds lower than the average of the three bee traits ($0.92\,C \pm 0.079\,eV$ for larval respiration, $1.06 \pm 0.28\,eV$ for pupal development, and $0.83 \pm 0.06\,eV$ for adult muscle force production (Fig. 2)). This parameter indicates the extent to which a trait is affected by changes in temperature, with higher values corresponding to stronger temperature-mediated effects. For example, whereas

viral protease activity varied <2-fold over a > 30 °C temperature range between 5 and 37.5 °C, larval respiration increased by 8-fold over a 20 °C range (18–38 °C), and adult muscle force production varied by two-fold over just a 10 °C interval (Fig. 1). The pupal stage used for injection is especially sensitive to temperature, with development only completing successfully between 29 and 37 °C (Fig. 1).

The temperature dependence of infection closely reflected that of pupal development, with a temperature-dependent 6-fold increase in infection from $7.66 \cdot 10^4 \pm 7.90 \cdot 10^3$ luminescence

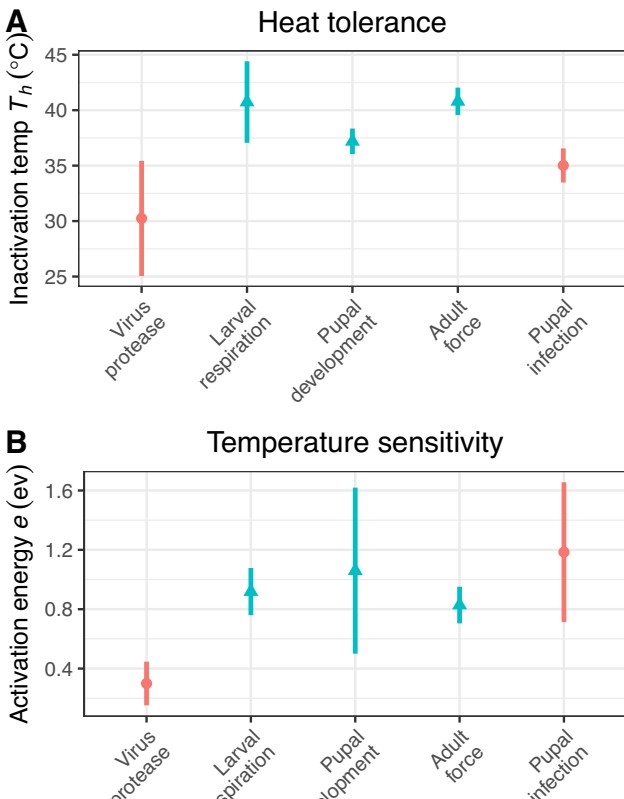

**Fig. 2 Heat tolerance and temperature responsiveness of virus protease and bee traits. A** Temperatures of 50% inactivation ($T_h$) and **B** activation energies ($e$) for temperature sensitivity of virus protease (red circles) and bee traits (blue triangles) from Sharpe–Schoolfield models depicted in Fig. 1. Points and error bars show model estimates ±2 standard errors.

units SE at 22 °C to $4.64 \cdot 10^5 \pm 4.54 \cdot 10^4$ units at 34 °C, then a sharp decline (by 98%) to $9.43 \cdot 10^3 \pm 2.91 \cdot 10^3$ SE at 37 °C (Fig. 1). This corresponds to an increase of over ten thousand-fold at the most permissive temperatures relative to the samples taken immediately post-injection (mean $11.9 \pm 3.0$ SD), indicating successful proliferation of the virus over the 48 h incubation period. The estimated temperature of 50% inactivation was well-defined ($35.0 \pm 0.76$ °C SE) and closest to that of pupae, although confidence intervals overlapped slightly with the broad estimate for the viral protease (Fig. 2). The high temperature sensitivity of infection was reflected by a high estimated activation energy ($e = 1.18 \pm 0.23$ eV SE) much closer to that of the bee traits than the activity of the viral protease (Fig. 2).

## Discussion

Our experiments with the DWV protease are qualitatively consistent with this protein's initial characterization[43], indicating a temperature sensitivity (activation energy $e < 0.3$ eV) at the low end of the usual range observed for biochemical reactions of metabolism (0.6–0.7, range 0.2–1.2 eV[44],). The maintenance of protease function at >50% of maximum activity over a > 30 °C range is consistent with infectivity in honey bees across a range of season- and host caste-related temperatures[23], and associations of the virus with a wide range of arthropod hosts that exhibit varying degrees of endothermy[36]. In contrast, the high (>35 °C)

temperatures of honey bee peak performance—well above the 25–29 °C reported for most insect traits[3]—and high thermal sensitivity of honey bee traits ($e > 0.9$ eV) are consistent with this species' endothermically controlled body temperature during key periods and activities (e.g., brood development, foraging)[45]. This control is exemplified by maintenance of temperatures within < 1 °C of 35 °C at the colony core[20], site of the highly temperature-sensitive pupal life stage.

Our thermal performance curves of the viral protease and honey bee traits identified a relatively large temperature window—spanning the typical range of colony temperatures—over which viral protease activity declined while honey bee performance rapidly increased. These contrasting responses of host and parasite to temperature allowed us to test the thermal mismatch hypothesis using pupal infection. Specifically, the higher temperature sensitivity and warmer temperatures of peak activity of bee traits relative to the virus enzyme suggested that the relative performance of parasites vs. hosts would decline steeply throughout the range of temperatures typical of the colony, resulting in a corresponding reduction in infection intensity. However, we observed the opposite result, with the temperature dependence of infection—which exhibited a high temperature sensitivity and a well-defined peak in the region of brood incubation temperatures—corresponding to host performance rather than to the relative performance of viruses and hosts. The breakdown in viral proliferation occurred only over the >34.5 °C range where pupal development also becomes threatened, as manifested by decreased emergence success, alteration of brain development, and disruption of subsequent behavior as adults[26,46].

Although this result contrasts with predictions based on the expected relative performance of hosts and parasites, it is consistent with a previous study that examined naturally occurring DWV infection in adult bees[33], in which a 37 °C rearing temperature resulted in 8- to 17-fold reductions in infection intensity relative to 28 and 34 °C. The latter temperatures also spanned the range of optimal adult survival, the 28 °C preferred honey bee sleeping temperature[25], and the 32 °C temperature in which honey bee thermoregulatory energy expenditure is minimized[47]. This concordance between the performance of hosts and viral infection could reflect the high host dependence of virus production, which exploits the resources and machinery of the host cell, resulting in viral growth rates that are limited by the efficiency of the host as a viral factory. Our results with DWV are also comparable to that of honey bee infection with *Nosema* spp. microsporidia, which like viruses are intracellular, host energy-dependent pathogens. Infection intensity was increased by 10-fold at 33 °C relative to 25 °C, then decreased by over two orders of magnitude at 37 °C[31], mirroring the pattern found here and possibly reflecting niche overlap between these host-dependent parasites, which appear to compete with one another in bees[48]. Infection with these types of parasites may be less well-predicted by the thermal mismatch hypothesis relative to other parasites that can reproduce independently of host cells[9], or whose proliferation is more effectively controlled by a temperature-dependent host immune response.

Despite this host-dependence of intracellular parasites, a match between the temperature dependence of host and parasite performance is certainly not a foregone conclusion; there are numerous counterexamples of arthropod-parasite systems where parasite proliferate fastest at temperatures that are not ideal for hosts. In honey bees, Bee virus X was reported to proliferate in young bees incubated at a below-optimal 30 °C, but not in those incubated at the 35 °C found in their colony[34], suggesting a negative relationship between host performance and infection. In mosquito vectors of mammal viruses, infection with most species

developed fastest at 37 °C, some 5 °C to 9 °C higher than the optimum temperatures for most of the mosquito traits[6]. Although these viruses are an exception to the pattern of low infectivity among ectotherm-hosted viruses above 35 °C[15,19] —a necessary adaptation for infection of the mammalian bloodstream—they provide additional examples of peak proliferation rates in temperature ranges where insect host physiology is compromised. Hence, their temperature-dependent development in the insect vector is consistent with the predictions of the thermal mismatch hypothesis for performance of a high temperature-adapted parasite in a less heat-tolerant host—i.e., the mammal-hosted viruses are expected to perform well at high, mammal-like temperatures, whereas the mosquito vectors are expected to resist infection most strongly at the lower (near-ambient) temperatures to which these insects are accustomed. Our finding of concordance between host and parasite performance also contrasts with that of another arthropod host-microsporidian parasite system (*Daphnia magna* and *Ordospora colligata*), in which parasite growth rate increased over a gradient of temperature that reduced host lifespan by 10-fold (among both parasite-exposed and control hosts)[4]. Examination of additional virus proteins and processes, and comparison of proliferation in identified hosts[36,37] or compatible cell lines[39] with different temperature sensitivities and thermal optima, should allow further evaluation of the relationship between host and virus performance and infection with this widespread virus, testing the generality of and examining the mechanisms underlying our result.

Regarding the potential for high colony temperatures to counteract DWV proliferation, although our finding of strongly reduced infection above 34 °C is consistent with high temperature-mediated reductions in insect-, plant-, and bacterium-infecting viruses[10,15,19,49,50], the 37 °C upper limit of pupal development suggests either a narrow thermal safety margin or a heavy cost of high temperature-based resistance for hosts. Even in adults, resistance to the virus came at the expense of host survival[33], implying that elevation of body temperature up to or beyond 37 °C would not be considered an adaptive "self-medication" response[51]. Nevertheless, although viral protease function is preserved at higher temperatures than can be tolerated by pupae, it is possible that other key elements of the virus life cycle are disrupted as temperatures approach 37 °C. As honey bees clearly can regulate temperatures between 35 and 37 °C with better than 1 °C precision[20,21,23], at least in the brood at the colony center, we cannot rule out a temperature window over which host health is preserved but virus replication is inhibited. It would be of interest to see whether virus inoculation leads to differences in thermal preference, as seen in bees exposed to the similarly heat-sensitive infection with *Nosema*[52].

DWV infects honey bees throughout the year as well as diverse arthropods—including some hosts where infection may be virulent[38,53]—that exhibit a range of life history strategies and thermoregulatory abilities[37]. By elucidating factors governing virus replication, including host temperature, this study system has great potential for testing the thermal mismatch hypothesis of host-parasite interactions. Such investigations can evaluate the immunological value of endothermy in predominantly ectothermic host-parasite communities, further elucidating parasite host range, seasonal dynamics, and spillover potential and consequences across seasons, climates, and host communities. This work may also add nuance to the thermal mismatch hypothesis by considering both the parasite-facilitating and -inhibiting aspects of host performance.

## Methods

**Temperature sensitivity of viral protease activity.** We tested the function of a recombinant DWV 3C protease[43] (see Supplementary Note 1 for full amino acid sequence) in a fluorescence resonance energy transfer (FRET) assay across temperatures from 5 to 50 °C. The protease (100 nM) was incubated with 80 µM of a fluorogenic DABCYL- VQAKPEMDNPNPG-EDANS-peptide substrate corresponding to the identified target peptide at the interface between the leader protein and VP2 proteins of DWV[54] with a buffer consisting of 50 mM Tris [pH 7.0], 150 mM NaCl, 1 mM EDTA, 2 mM DTT, and 10% glycerol (see Supplementary Note 1 for peptide amino acid sequence and reaction buffer composition). The assay was run in temperature-controlled thermocyclers with 50 µL of reaction mixture per well in a standard PCR plate.

We conducted three runs with a temperature range of 5–30 °C (in 5 °C increments) and one run from 15 to 30 °C (in 3 °C increments) using parallel incubation in six thermocyclers, and three runs of eight temperatures each spanning 30–50 °C using the thermocycler's temperature gradient feature, yielding 48 observations of reaction rate over a 25 °C temperature range. The reaction was sampled at 0, 15, and 30 min of incubation (excluding an initial 5-min equilibration period) by transferring two aliquots per temperature (pooled volume: 80 µL) to a 96-well assay plate, which was read immediately in a Biotek Synergy H1 spectrophotometer (excitation: 340 nm, emission: 520 nm). The maximum rate of increase in fluorescence over either 15-min period was calculated for each sample ($n = 1$ rate per temperature per run). Net reaction rate was computed by subtracting the maximum rate of increase of a control sample incubated in parallel at the corresponding temperature in the absence of protease, to control for possible differences in spontaneous fluorescence of the substrate across temperatures.

**Temperature sensitivity of honey bee traits.** To compare the temperature sensitivity of the virus protease to that of bee traits, we gathered data for larval respiration (18–38 °C), pupal development rate (28–38 °C), and adult muscle force production during flight (thoracic temperatures 27.5–47 °C) from existing literature[24,26,45]. Briefly, larval respiration was measured as $CO_2$ production (µL mg$^{-1}$ h$^{-1}$) of 3.5-day-old larvae over a 20 min time interval in 4 °C increments from 18 to 38 °C[45]. Pupal development rate (d$^{-1}$) was measured by incubating combs of freshly capped brood in 1 °C increments from 28 to 38 °C and recording the number of days until adult bees emerged[26]. Adult force production (mN) was recorded from flight muscles of tethered bees at 7 thoracic temperatures from 27.5–47 °C[24]. We elected not to focus on adult respiration rates because even in isolation, bees may be endothermic at low temperatures and become agitated at high temperature, making it difficult to estimate true resting metabolic rates[55].

**Temperature sensitivity of virus infection in honey bee pupae.** To examine the temperature sensitivity of pupal infection, we used a nanoluciferase-tagged reporter clone of DWV previously developed in our laboratory[42]. Infection with this reporter clone can be rapidly quantified by luminescence—which has a nearly 1:1 correlation with virus copy number over four orders of magnitude (slope = 0.98, r$^2$ = 0.96)—using a spectrophotometer[42]. Purple-eyed pupae were removed from brood frames using scalpel and forceps. We injected 32 pupae per temperature with 10$^7$ virus particles of reporter-tagged virus-containing inoculum in 4.0 µL of sterile phosphate buffer saline (PBS). Pupae were incubated at six temperatures from 22 to 37 °C (3 °C increments). We chose this temperature range because it spans the typical temperature range experienced by bee brood and adults[21] and corresponds to the region where increases in temperature have opposite effects on viral protease activity and bee performance (i.e., protease activity decreases while bee performance increases), suggesting that small changes in temperature could have large and biologically relevant effects on infection.

Injected pupae were incubated for 48 h on filter paper in covered petri dishes. Incubators were manually humidified by placing shallow, water-filled containers on the bottom. After 48 h, pupae were frozen at −80 °C. For luciferase activity-based quantification of infection intensity, samples were individually homogenized for 30 s with glass microbeads in 100 µL PBS. A 5 µL aliquot of homogenate was diluted 10-fold in PBS in a black 96-well microplate. To each sample, we added 50 µL of 2x luciferase substrate-containing buffer from a commercial kit (Promega, Madison, WI). Total luminescence was measured immediately using a spectrophotometer (100 ms integration time). The raw luminescence value was used as the response variable to model the effects of temperature on infection intensity. Sixteen samples frozen immediately post-injection were assayed in parallel to estimate virus replication.

**Estimation of thermal performance curves.** The effects of temperature on each trait (i.e., protease activity, each bee trait (larval respiration, pupal development, and adult force production), and pupal infection) were modeled using a Sharpe–Schoolfield equation modified for high temperatures[7,10,56].

$$\text{rate(T)} = \frac{r_{T_{ref}} \cdot \exp^{\frac{-e}{k}\left(\frac{1}{T} - \frac{1}{T_{ref}}\right)}}{1 + \exp^{\frac{e_h}{k}\left(\frac{1}{T_h} - \frac{1}{T}\right)}} \tag{1}$$

In Eq. (1), *rate* refers to the observed response variable; $r_{T_{ref}}$ is the rate at the calibration temperature $T_{ref}$ (293 K, i.e., 20 °C); $e$ is the activation energy (in eV), which primarily affects the upward slope of the thermal performance curve (i.e., sensitivity of the trait to temperature) at suboptimal temperatures; $k$ is Boltzmann's

constant ($8.62 \cdot 10^{-5}$ eV·K$^{-1}$); $e_h$ is the deactivation energy (in eV), which determines how rapidly the thermal performance curve decreases at temperatures above the temperature of peak rate; $T_h$ is the high temperature (in K) at which rate is reduced by 50% (relative to the value predicted by the Arrhenius equation—which assumes a monotonic, temperature-dependent increase);[7] and $T$ is the experimental incubation temperature (in K). Models were fit using the *rTPC* package of R[57,58]. Confidence intervals on parameter values were defined as the range ±2 standard errors from the model estimate; estimates were considered significantly different when these intervals did not overlap for different traits. We focus our discussion on the parameters $e$ and $T_h$, as these were deemed most biologically relevant to this system and could be accurately estimated for all traits given the range of temperatures for which data were available. Because we measured a single virus trait, we could not conduct formal statistical tests to compare parameter estimates between bees and viruses; however, we highlight instances where 95% confidence intervals were non-overlapping in virus and host. Confidence intervals on predicted rates were obtained by bootstrap resampling of the model residuals (10,000 model iterations) using R package "car"[59]. To display the temperature responsiveness of all traits on the same scale, we scaled each thermal performance curve to the trait's maximum model-predicted value to yield a proportion of peak activity at each temperature.

**Statistics and reproducibility.** Numbers of replicates are described in the details of each experiment. All computed statistics are described in "Methods: Estimation of thermal performance curves".

**Reporting summary**. Further information on research design is available in the Nature Portfolio Reporting Summary linked to this article.

## Data availability
All data are supplied in the Supplementary Information, Supplementary Data 1.

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

## Acknowledgements

This project was supported by the USDA Agricultural Research Service Beltsville Bee Research Laboratory in house fund; USDA-NIFA Pollinator Health Grant 2020-67013-31861 to J.D.E.; and a North American Pollinator Protection Campaign Honey Bee Health Improvement Project Grant and an Eva Crane Trust Grant to E.C.P.Y. and J.D.E. Funders had no role in study design, data collection and interpretation, or publication. We thank the reviewers for their service in improving the manuscript.

## Author contributions

E.C.P.Y., E.V.R., and J.D.E. conceived the study. All authors (E.C.P.Y., E.V.R., L.M.M., D.L.B., K.G., A.P., R.P., J.D.E.) conducted experiments. E.C.P.Y. analyzed data and drafted the manuscript. All authors (E.C.P.Y., E.V.R., L.M.M., D.L.B., K.G., A.P., R.P., J.D.E.) revised the manuscript and gave approval for publication.

## Competing interests

All authors declare no competing interests.
