## [Peer Review File · Communications Biology]

Reviewers' comments:

Reviewer #1 (Remarks to the Author):

In this paper the authors investigated the temperature-sensitivity of DWV and its host (honey bees). This is a highly relevant topic for better understanding the dynamics of viruses in bees. They found that the virus performances are temperature sensitive but highly dependent on the host physiology. The paper is well written and the analysis are clear and relevant; I want to congratulate the authors for that. I have only some minor comments.

- The experiments were focused on DWV, so maybe it needs to be indicated in the title, especially since in line 264, they mention that the performance of the Bee virus X is not really dependent on the host but rather on the temperature (solely). So, the host-driven temperature dependence might not apply to all honey bee viruses.

In addition, the host-driven temperature dependence has been determined in pupae, but it is not clear if it also applies to adult bees. Maybe it should be specified in the title (honey bee pupae?)

- Lines 32-35: can you split the sentence in two, it is a bit difficult to read

- The next sentence is also a bit difficult to understand: is there a way to better explain the 'thermal mismatch hypothesis'?

- Line 55: it seems that all described pathogens have been linked to colony collapse. It doesn't think this is the case. They might affect bees at the individual level, but I am not sure they all have been related to colony collapse.

- Line 77: what does it mean 'that mature at ambient temperature'? Can you be more specific? You mean adult bees?

- Is it accepted to speak of physiology for viruses? Can we state that viruses have a specific physiology?

- Lines 128-133: I understand the data were retrieved from previously published studies. But it would be nice to have more details on the type of bees (young, nurse, forager, age) and the overall protocols/methods.

- Line 172: Parameter e (lower case? Like in the formula)

- Line 177: lower than what?

- Protease's activation energy: it would be helpful to have in the methods or results section an explanation of what it means, how do you interpret it?

- Fig. 1C: There is a strong increase in virus infection level at $\sim 28^{\circ}\text{C}$ (before the optimal brood temperature), and it kind of corresponds to the optimum for the virus protease: it could be discussed. In addition, the temperature dependence of the virus protease might better reflect a temperature dependence of virus infection level in adult bees which can tolerate a larger range of temperatures than larva and pupae. There might be strong dependence on host function at the immature stage but less at the adult stage. This could be also discussed.

- Fig. 2: can you perform statistical analysis to compare T_h and e_v values across the virus and bee traits?

Reviewer #2 (Remarks to the Author):

Temperature can have a strong effect on organism performance. In host-parasite interactions, the thermal mismatch hypothesis offers an explanation for how exposure to temperatures outside the adapted range of either the host or parasite affects their respective performance, ultimately shaping infection outcomes. For example, host-generated fever may expose parasites to temperatures above performance limits. Here, the authors test the thermal mismatch hypothesis through the interaction of honey bees, an endothermic social insect, and a prevalent hive parasite, Deformed wing virus (DWV). The authors compared host and virus performance over a range of temperatures, especially those relevant to honey bees. Activity of isolated DWV protease was quantified at different temperatures as

a proxy for virus performance. Measurements of honey bee larval respiration, pupal development, and adult muscle force were extracted from published reports to construct the host response to different temperatures. Results showed that DWV protease 'performed optimally' through a relatively large range of temperatures that were below those optimal for host performance, while host traits peaked at temperatures where viral protease was in decline. Contrary to the thermal mismatch hypothesis, infection of pupae with a luciferase-tagged DWV clone showed that infection intensity correlated with pupal performance at different temperatures. The ex-situ performance of DWV protease would suggest higher infection intensities at suboptimal honey bees temperatures, supporting the thermal mismatch hypothesis. However, the infection data and corresponding estimates of temperature sensitivity and activation energy suggest DWV's dependence on host resources for replication links its success to optimal host temperature. The following comments are offered:

1. The work presents results that are of interest to apicultural research given the worldwide prevalence of honey bee viruses, with virulence of some driven partially by association with the mite parasite *Varroa destructor*, as well as a broader audience in parasitology seeking to further understanding on factors that shape host-parasite interactions.
2. More description could be given for how individual viral and host traits were transformed to proportions. If appealing to a broader audience, it may also be good to include, briefly, experimental conditions/context of Petz et al. (larval respiration), Groh et al. (pupal development), Harrison and Fewell (adult muscle activity) in Methods (Lines 128-133).
3. Figure 1, panel 3 (Pupal infection). The scale of the y-axis is somewhat confusing based on the distribution of the individual data points relative to the model trend line. Would it be clearer to add a second y-axis on the right side of the figure that conveys infection intensity on a luminescence scale?
4. There appears to be a distinct separation between groups of pupae with high intensity and pupae with low intensity; infections are largely absent from the "middle ground" of luminescence at several temperature points (Figure 1, panel 3). The model appears to accommodate this absence with an estimate that lies between the low and high clusters. Could the authors provide an explanation for the absence of mid-range infections given the placement of the estimate and the seemingly narrow confidence interval? The estimate may not reflect the "bimodal" distribution of the data.
5. Referring to Lines 202-204, would it be useful to add a reference point/line on Figure 1, panel 3 that shows the location of luminescence taken immediately post-injection? Were sham-infected bees included as controls and checked for background luminescence?
6. Figure 2 should be introduced earlier, such as the first paragraph of Results, or activation energy and temperature tolerance results moved to the second paragraph of Results.
7. Lines 267-271: Since the mosquito is the vector and not the host, the performance of viruses would be more reflective of optimal host performance and not vector performance? Not sure if this is a good example for supporting the mismatch hypothesis.
8. It is fitting that the authors comment on limitations of examining isolated DWV protease and need for further evaluation of the response of viral and host proteins/processes to different temperatures in whole systems to further characterize the response of viral infection to host-relevant temperatures.

We are grateful for the constructive comments offered by the Reviewers, whose points we address in this response letter and incorporate into the revised manuscript.

We have made the following major changes:

- Revised the title to indicate the study system, including virus and bee life stage investigated
- Added details of the prior studies on bee traits incorporated into the analysis
- Better explained the details of our analyses
- Revised the figure on pupal infection to display the initial infection intensity and successful replication of the virus.

We provide detailed responses to each comment below.

Reviewer #1

I have only some minor comments.

- The experiments were focused on DWV, so maybe it needs to be indicated in the title, especially since in line 264, they mention that the performance of the Bee virus X is not really dependent on the host but rather on the temperature (solely). So, the host-driven temperature dependence might not apply to all honey bee viruses. In addition, the host-driven temperature dependence has been determined in pupae, but it is not clear if it also apply to adult bees. Maybe it should be specify in the title (honey bee pupae?)

Response: We agree and have revised the title to indicate both the virus and life stage tested. The new title is: "Host-driven temperature dependence of Deformed wing virus infection in honey bee pupae".

- Lines 32-35: can you split the sentence in two, it is a bit difficult to read

Response: We split the sentence into two sentences as suggested.

Metabolic theory provides a framework for explaining the temperature dependence of physiological and ecological phenomena, including host-parasite interactions³⁻⁵. This approach uses models derived from enzyme kinetics to describe the temperature dependence of organismal performance (i.e., the 'thermal performance curve')^{6,7}. **(beginning Line 33)**

- The next sentence is also a bit difficult to understand: is there a way to better explain the 'thermal mismatch hypothesis'?

Response: We apologize for the difficulty and have now provided a more thorough explanation of the hypothesis and its predictions.

(Under the thermal mismatch hypothesis)... temperature is predicted to impact infection when hosts and parasites respond differently or, in extreme cases, oppositely, over a given temperature range, with parasite growth maximized at temperatures where parasites outperform hosts and minimized where hosts outperform parasites^{4,9,10}. In general, hosts are expected to be most resistant to infection at temperatures within the preferred host range, resulting in lower infection of warm-adapted hosts at warm temperatures... **(beginning Line 38)**

- Line 55: *it seems that all described pathogens have been linked to colony collapse. It don't think this is the case. They might affect bees at the individual level, but I am not sure they all have been related to colony collapse.*

Response: Fair enough. We have adopted more conservative wording, and now simply say that honey bees face an 'assortment of parasites and pathogens, which can adversely affect individual and colony health.' **(Line 60)**

- Line 77: *what does it mean 'that mature at ambient temperature'? Can you be more specific? You mean adult bees?*

Response: We have clarified that this sentence refers to ectothermic hosts:

(Relative to honey bees,) much greater temperature ranges occur in DWV-associated, non-honey bee host species that do not thermoregulate during development. **(Line 83)**

- *Is it accepted to speak of physiology for viruses? Can we state that viruses have a specific physiology?*

Response: Yes! As the virus is part of a living organism with a genome, physical structure, and biochemistry, it is accepted to speak of 'virus physiology', which is considered the subfield of virology that deals with the processes and activities of viruses. However, to avoid any potential confusion, we have replaced 'physiology' with 'functions' or 'traits' when referring to the virus in the text.

- *Lines 128-133: I understand the data were retrieved from previously published studies. But it would be nice to have more details on the type of bees (young, nurse, forager, age) and the overall protocols/methods.*

Response: We agree that it could be helpful for the reader to have some context for these measurements. We have added a few details of their experiments to the Methods:

Briefly, larval respiration was measured by CO₂ production ($\mu\text{L mg}^{-1} \text{h}^{-1}$) of 3.5-day-old larvae over a 20 min time interval in 4 °C increments from 18 to 38 °C⁴⁵. Pupal development rate (d^{-1}) was measured by incubating combs of freshly capped brood in 1 °C increments from 28 to 38 °C and recording the number of days until adult bees emerged²⁶. Adult force production (mN) was recorded from flight muscles of tethered bees at 7 thoracic temperatures from 27.5-47 °C²⁴.
(beginning Line 135)

- *Line 172: Parameter e (lower case? Like in the formula)*

Response: Good catch! We have changed *E* to *e*.

- *Line 177: lower than what?*

Response: Sorry for the confusion. We rephrased the text to make it clear that this sentence compares the virus's traits to those of the bee host:

Relative to the bee traits, activity of the viral protease peaked at lower temperatures but was overall less temperature-sensitive. **(Line 191)**

- *Protease's activation energy: it would be helpful to have in the methods or results section an explanation of what it means, how do you interpret it?*

Response: We agree that it is helpful to explain this parameter within the context of the results. Accordingly, we added the sentence:

This parameter (e) indicates the extent to which a trait is affected by changes in temperature, with higher values corresponding to stronger temperature-mediated effects. **(Line 199)**

This supplements the more technical definition of the parameter in the Methods:

e is the activation energy (in eV), which primarily affects the upward slope of the thermal performance curve (i.e., sensitivity of the trait to temperature) at suboptimal temperatures. **(Line 171)**

- Fig. 1C: There is a strong increase in virus infection level at ~28°C (before the optimal brood temperature), and it kind of corresponds to the optimum for the virus protease: it could be discussed. In addition, the temperature dependence of the virus protease might better reflect a temperature dependence of virus infection level in adult bees which can tolerate a larger range of temperatures than larva and pupae. There might be strong dependence on host function at the immature stage but less at the adult stage. This could be also discussed.

Response: We agree that the model for the virus protease indicates near-maximal activity at 28 °C. However, there is minimal (<10%) variation between 22 and 28 °C, with no obvious peak in the thermal performance curve (Fig. 1, upper panel). This contrasts with the narrow, markedly high-temperature peaked curve for pupal infection (Fig. 1, lower panel), which bears uncanny resemblance to the larval and pupal curves in Fig. 1 (middle panel). In the Discussion, we explain that:

(We observed a) temperature dependence of infection—which exhibited a high temperature sensitivity and a well-defined peak in the region of brood incubation temperatures—corresponding to host performance rather than to the relative performance of viruses and hosts. **(beginning Line 259)**

Regarding the temperature dependence of infection in adults, we have discussed that the existing literature suggests similar host-dependent viral proliferation in this life stage:

(Our result in pupae) is consistent with a previous study that examined naturally occurring DWV infection in adult bees³³, in which a 37 °C rearing temperature resulted in 8- to 17-fold reductions in infection intensity relative to 28 and 34 °C. The latter temperatures also spanned the range of optimal adult survival, the 28 °C preferred honey bee sleeping temperature²⁵, and the 32 °C temperature in which honey bee thermoregulatory energy expenditure is minimized⁵³. This concordance between the performance of hosts and viral infection could reflect the high host dependence of virus production. **(beginning Line 267)**

We also discuss the broad temperature tolerance of viral enzyme function as a possible adaptation to infection of honey bees and other hosts that tolerate a larger range of temperatures than do honey bee larvae and pupae:

The maintenance of protease function at >50% of maximum activity over a >30 °C range is consistent with infectivity in honey bees across a range of seasonally and host caste-related temperatures²³ and association with a wide range of arthropod hosts that exhibit varying degrees of endothermy³⁶. **(beginning Line 243)**

- Fig. 2: can you perform statistical analysis to compare T_h and e_v values across the virus and bee traits?

Response: We explain that our analysis is focused on separation of confidence intervals rather than traditional statistical tests, as we technically had $n = 1$ trait estimate for the virus:

Because we measured a single virus trait, we could not conduct formal statistical tests to compare parameter estimates between bees and viruses; however we highlight instances where 95% confidence intervals were non-overlapping in virus and host. **(beginning Line 182)**

Reviewer #2

1. *The work presents results that are of interest to apicultural research given the worldwide prevalence of honey bee viruses, with virulence of some driven partially by association with the mite parasite Varroa destructor, as well as a broader audience in parasitology seeking to further understanding on factors that shape host-parasite interactions.*

Response: We appreciate the positive comments.

2. *More description could be given for how individual viral and host traits were transformed to proportions. If appealing to a broader audience, it may also be good to include, briefly, experimental conditions/context of Petz et al. (larval respiration), Groh et al. (pupal development), Harrison and Fewell (adult muscle activity) in Methods (Lines 128-133).*

Response: We have provided more information on these prior studies, as described in response to the comment from Reviewer 1.

We have also provided an explanation of the transformation of thermal performance curves to proportions:

To display the temperature responsiveness of all traits on the same scale, we scaled each thermal performance curve to the trait's maximum model-predicted value to yield a proportion of peak activity at each temperature. **(beginning Line 186)**

We also explain this conversion in the Figure 1 legend:

To provide a comparable scale for the different measurements, raw values for all traits are shown as proportions of the trait's peak model-predicted value. **(Line 217)**

3. *Figure 1, panel 3 (Pupal infection). The scale of the y-axis is somewhat confusing based on the distribution of the individual data points relative to the model trend line. Would it be clearer to add a second y-axis on the right side of the figure that conveys infection intensity on a luminescence scale?*

Response: As detailed in response to the previous comment, we chose to graph each trait on a proportional scale (i.e., relative to the trait's peak model-predicted value) to facilitate cross-trait comparisons. The scale for the absolute luminescence measurements is largely arbitrary, reflecting the signal amplification, exposure duration, and luminometer sensitivity, so we consider it most meaningful to show the data on a relative scale in the figures. But we have now added a secondary y-axis to the

lower panel of Figure 1 to show raw luminescence, in addition to a horizontal reference line for the Time 0 samples (as suggested in Comment 5).

4. *There appears to be a distinct separation between groups of pupae with high intensity and pupae with low intensity; infections are largely absent from the “middle ground” of luminescence at several temperature points (Figure 1, panel 3). The model appears to accommodate this absence with an estimate that lies between the low and high clusters. Could the authors provide an explanation for the absence of mid-range infections given the placement of the estimate and the seemingly narrow confidence interval? The estimate may not reflect the “bimodal” distribution of the data.*

Response: Yes, there appears to be a strong separation between lightly and heavily infected pupae. This is a pattern we have observed before in DWV-injected pupae (e.g., Ryabov *et al.* 2019 PLOS Biology, <https://doi.org/10.1371/journal.pbio.3000502>). We understand the concern that there are few data points near the model-predicted values for some temperatures. However, the model estimates mean infection at each temperature. The result here is analogous to a logistic model on a binary response variable—although each individual observation is a 0 or 1, all the model-predicted values are proportions that lie between these two possible outcomes. The shaded band represents a confidence interval for the mean, not a prediction interval for individual observations. Because it is constructed by (non-parametric) bootstrapping, it should be relatively free of distributional assumptions, with the width of the band positively related to unexplained variation but inversely related to sample size. We highlight the bootstrap procedure in both the Methods and the Figure 1 legend.

Methods:

Confidence intervals on predicted rates were obtained by bootstrap resampling of the model residuals (10,000 model iterations) using R package “car” (**Line 185**)

Figure 1 legend:

Points show individual observations; lines and shaded bands show model predictions and 95% bootstrap confidence intervals. (**Line 210**)

5. *Referring to Lines 202-204, would it be useful to add a reference point/line on Figure 1, panel 3 that shows the location of luminescence taken immediately post-injection? Were sham-infected bees included as controls and checked for background luminescence?*

Response: Good idea. We have added a horizontal reference line to the Figure 1, panel 3 to show post-injection luminescence, in addition to the secondary y-axis suggested in Comment 3.

The new figure and text added to the Figure 1 legend are duplicated here for easy reference:

Trait ● Larval respiration ▲ Pupal development ■ Adult muscle force

In the lower panel, the secondary y-axis indicates the raw sample luminescence (i.e., before transformation to proportion of maximum intensity) and the horizontal line dotted line represents the mean luminescence of $n = 16$ samples frozen immediately post-injection. **(beginning Line 214)**

We also noted the initial values and compared them to the final readings in the Results:

(Infection intensity at the most permissive temperature) corresponds to an increase of over ten thousand-fold at the most permissive temperatures relative to the samples taken immediately post-injection (mean 11.9 ± 3.0 SD), indicating successful proliferation of the virus over the 48 h incubation period. **(beginning Line 223)**

6. Figure 2 should be introduced earlier, such as the first paragraph of Results, or activation energy and temperature tolerance results moved to the second paragraph of Results.

Response: We agree and now reference Figure 2 twice in the first paragraph of the Results **(beginning Line 191)**.

7. Lines 267-271: Since the mosquito is the vector and not the host, the performance of viruses would be more reflective of optimal host performance and not vector performance? Not sure if this is a good example for supporting the mismatch hypothesis.

Response: We completely agree with the Reviewer's assertion that the virus's infection of the mosquito represents an instance of an endothermic host-adapted parasite developing in an ectothermic host; we have edited the text to clarify this. The mismatch hypothesis postulates that changes in the success of infection reflect the relative performance of hosts and parasites across a temperature range (not that the parasite itself performs best in a non-host temperature range). In this case:

(The mammalian viruses') temperature-dependent development in the insect vector is consistent with the predictions of the thermal mismatch hypothesis for performance of a high temperature-adapted parasite in a less heat-tolerant host—i.e., the mammal-hosted viruses are expected to perform well at high, mammal-like temperatures, whereas the mosquito vectors are expected to resist infection most strongly at the lower (near-ambient) temperatures to which these insects are accustomed. **(beginning Line 293)**

8. It is fitting that the authors comment on limitations of examining isolated DWV protease and need for further evaluation of the response of viral and host proteins/processes to different temperatures in whole systems to further characterize the response of viral infection to host-relevant temperatures.

Response: We appreciate the Reviewer's support for our conservative stance.